# Angiotensin-Converting Enzyme 2 (ACE2) as a Potential Diagnostic and Prognostic Biomarker for Chronic Inflammatory Lung Diseases

**DOI:** 10.3390/genes12071054

**Published:** 2021-07-09

**Authors:** Dejan Marčetić, Miroslav Samaržija, Andrea Vukić Dugac, Jelena Knežević

**Affiliations:** 1Department of Internal and Pulmonary Diseases, General Hospital Virovitica, Ljudevita Gaja 21, 33000 Virovitica, Croatia; marcetic.pulmo@gmail.com; 2Department of Lung Diseases Jordanovac, Zagreb University Hospital Centre, School of Medicine, University of Zagreb, 10000 Zagreb, Croatia; miroslav.samarzija@kbc-zagreb.hr (M.S.); avukic@kbc-zagreb.hr (A.V.D.); 3Laboratory for Advanced Genomics, Division of Molecular Medicine, Ruđer Bošković Institute, 10000 Zagreb, Croatia; 4Faculty of Dental Medicine and Health, Josip Juraj Strossmayer University of Osijek, 31000 Osijek, Croatia

**Keywords:** angiotensin-converting enzyme 2 (ACE2), renin-angiotensin-aldosterone system (RAAS), chronic inflammatory lung diseases

## Abstract

Chronic inflammatory lung diseases are characterized by uncontrolled immune response in the airways as their main pathophysiological manifestation. The lack of specific diagnostic and therapeutic biomarkers for many pulmonary diseases represents a major challenge for pulmonologists. The majority of the currently approved therapeutic approaches are focused on achieving disease remission, although there is no guarantee of complete recovery. It is known that angiotensin-converting enzyme 2 (ACE2), an important counter-regulatory component of the renin–angiotensin–aldosterone system (RAAS), is expressed in the airways. It has been shown that ACE2 plays a role in systemic regulation of the cardiovascular and renal systems, lungs and liver by acting on blood pressure, electrolyte balance control mechanisms and inflammation. Its protective role in the lungs has also been presented, but the exact pathophysiological mechanism of action is still elusive. The aim of this study is to review and discuss recent findings about ACE2, including its potential role in the pathophysiology of chronic inflammatory lung diseases:, i.e., chronic obstructive pulmonary disease, asthma, and pulmonary hypertension. Additionally, in the light of the coronavirus 2019 disease (COVID-19), we will discuss the role of ACE2 in the pathophysiology of this disease, mainly represented by different grades of pulmonary problems. We believe that these insights will open up new perspectives for the future use of ACE2 as a potential biomarker for early diagnosis and monitoring of chronic inflammatory lung diseases.

## 1. Introduction

Angiotensin-converting enzyme 2 (ACE2) is a transmembrane glycoprotein discovered in the year 2000 [1,2]. *ACE2* gene is located on the X chromosome (cytogenetic location: Xp22.2) and consists of 18 exons that encode for protein of 805 amino acids. ACE2 is a type 1 integral membrane glycoprotein with two domains, the amino-terminal catalytic domain and carboxy-terminal transmembrane domain. The active domain of ACE2 is exposed to the extracellular surface, facilitating the metabolism of circulating peptides [1,2]. ACE2 is constitutively expressed by epithelial cells of the lungs—more precisely, on the surface of type I and type II alveolar epithelial cells [3]. ACE2 is also expressed in the vascular system—endothelial cells, migratory angiogenic cells, and vascular smooth muscle cells. In the heart, ACE2 is expressed in the cardiomyocytes, cardiac fibroblasts, coronary vascular endothelium and epicardial adipose tissue. In the kidneys, ACE2 was detected in glomerular endothelial cells, podocytes and proximal tubule epithelial cells. ACE2 is also expressed and functional in the liver, enterocytes of the intestines, and the central nervous system [4]. ACE2 is a component of the renin—angiotensin—aldosterone system (RAAS), a hormone system important in the regulation of blood pressure, fluid and electrolyte balance and the regulation of the systemic circulation [5]. Abnormal activation of the RAAS has been associated with the pathogenesis of hypertension, heart failure and renal diseases. Its involvement in the inflammation pathogenesis is also well known [6]. 

### 1.1. Physiological Function and Signaling Pathway of RAAS

The main physiological function of the RAAS is to regulate the cardiovascular system by controlling blood volume and blood tone during renal hypoperfusion. In addition to the systemic RAAS, there is also the tissue-specific RAAS, which both function independently of each other, and of the circulating RAAS. The tissue (local) RAAS has an important role in the pathogenesis of atherosclerosis, cardiac hypertrophy, type 2 diabetes and renal fibrosis [7]. Renin, angiotensin II and aldosterone play an important role in RAAS homeostasis. Renin is the initial protein in the RAAS signaling pathway. Renin is the proteolytic enzyme, secreted from the juxtaglomerular cells of the kidney as a response to a reduced amount of blood flow, sympathetic nerve stimulation, or activation by macula densa cells in response to decreased sodium in the distal tubule [8]. Upon activation, renin hydrolyzes angiotensinogen, a serum globulin produced in the liver, into angiotensin I (Ang I). Subsequently, Ang I is converted into angiotensin II (Ang II) via angiotensin-converting enzyme (ACE). Ang II has a powerful vasoconstriction effect (Figure 1). Ang II has effects on the arterioles, brain, adrenal cortex and kidney through two G-protein-coupled receptors, the angiotensin II type I (AT1R) and type II (AT2R) receptors. Ang II, a key RAAS peptide, has many regulatory roles. The binding of Ang II on the AT1R triggers vasoconstriction with an increase in blood pressure, inflammation, apoptosis and fibrosis, while binding on AT2R has opposite effects. The next step in RAAS signaling is Ang II conversion into angiotensin 1-7 (Ang 1-7), via ACE2. By binding to G-protein-coupled receptor Mas (Mas R), Ang 1-7 increases vasodilation and has an anti-inflammatory effect, opposite to that of Ang II [9]. 

### 1.2. ACE2 and RAAS Regulation

ACE2 has different roles ranging from a negative regulator of the renin–angiotensin system (peptidase-dependent) to an amino acid transporter and as a functional receptor for severe acute respiratory syndrome coronavirus (SARS-CoV) (peptidase-independent). ACE2 has an important role in the later stages of the RAAS cascade system by neutralizing the effect of accumulated Ang II. Ang II is hydrolyzed by ACE2 into the active product Ang 1-7, which interacts with Mas R. Binding of Ang 1-7 on Mas R promotes vasodilatation and decreases cell proliferation (Figure 2) [5,10]. It is known that in the lung, Ang II is able to induce bronchoconstriction, vasoconstriction, fibroproliferation, cytokine expression, and cell apoptosis, thus promoting tissue injury [11]. Therefore, in the respiratory system, ACE2 protects from lung injury [12], and it is possible that downregulation of ACE2 by SARS-CoV infection participates in the development of severe lung failure in SARS. 

RAAS could be activated/regulated in two opposite pathways, depending on physiological conditions. The classical pathway, formed by ACE/Ang II/AT1R, is associated with vasoconstriction, cell proliferation, organ hypertrophy, sodium retention and aldosterone release [13]. By the binding of Ang II to AT1R, the inflammatory cascade will be triggered and the NF-κB, STATs and MAPK pathways will be activated, stimulating the production of reactive oxygen species (ROS) and inducing apoptosis [14,15,16,17]. Another counter-regulatory, or vasodilator, pathway comprises the ACE2/Ang1-7/MasR axis, which is involved in vasodilation, anti-proliferation, anti-hypertrophy, cardio-protective and renoprotective actions [18]. Accordingly, the consequences of the ACE-AngII-AT1R axis could be reduced by activating the antagonistic ACE2/Ang1-7/MasR pathway.

### 1.3. ACE Inhibitors and ARBs

As has already been mentioned, the RAAS functions through the ACE/AngII/AT1 and ACE2/AngI (1-7)/MasR axes with opposite effects [19,20]. Due to their effects on these signaling pathways, ACE inhibitors and angiotensin II receptor blockers (ARBs) are widely used in the treatment of arterial hypertension and heart failure. They inhibit the conversion of Ang I into Ang II and block the binding of Ang II to AT1R, respectively. It is worth mentioning that the blocking of the AT1R has been shown to have a favorable effect in the pathogenesis of acute lung injury [21,22]. The antihypertensive effects of RAAS blockade are also partly determined by the ability of both ACE inhibitors and ARBs to increase the circulating levels of Ang 1-7 [23]. Ang II increases sympathetic activity and enhances water retention in renal tubules, resulting in the increase in the systemic blood pressure [24]. In arterioles, Ang II binds to AT1R, leading to a cascade that results in potent arteriolar vasoconstriction. Ang II also promotes the secretion of antidiuretic hormone (ADH) by the posterior, which acts to the increase water reabsorption in the kidney. By binding to the hypothalamus, Ang II stimulates the feeling of thirst and increases water intake. By acting on the adrenal cortex, Ang II stimulates the release of aldosterone, a steroid hormone that causes an increase in sodium reabsorption, leading to increased osmolarity and volume of extracellular liquid (Figure 1) [25]. The excess of aldosterone results in hypokalemia and muscle weakness, while deficiency results in hyperkalemia and toxic effects on the heart [26].

## 2. ACE2, Inflammation and Pulmonary Disease

Inflammation is the natural defense mechanism associated with many acute and chronic inflammatory diseases, including pulmonary diseases. It is a biological response of the immune system that can be triggered by a variety of factors. These factors include toxic compounds, pathogens and damaged cells. In the background of inflammation pathophysiology is the recruitment of the immune cells into the affected tissue aiming to remove the “danger signals”, such as pathogenic microorganism, irritants or dead cell debris. Most agents associated with chronic inflammation cause insidious but progressive and often extensive tissue necrosis accompanied by ongoing repair and fibrosis. The amount of fibrosis in the tissues is a function of the duration of chronic inflammation [27]. Chronic pulmonary diseases pose a significant public health problem, because their incidence is rising daily. It is well established that inflammation plays a key role in chronic pulmonary diseases, but the role of ACE2 in the pathophysiology behind these diseases is still not fully understood and somewhat contradictory [28,29]. It has been shown that the role of ACE2 in the pathogenesis of both chronic and acute inflammatory lung disease could be associated with promoting the ACE2/Ang1-7/MasR signaling pathway or the opposite ACE/AngII/AT1R pathway. Imai et al. demonstrated that ACE2 is protective in the mouse model of acute lung injury (ALI) induced by acid aspiration and sepsis. They showed that ACE2 knockout mice showed very severe disease when compared to wild-type mice. Loss of ACE2 expression in mutant mice resulted in enhanced vascular permeability, neutrophil accumulation, increased lung edema, and worsened lung function [20]. Additionally, in lung injury models induced by bleomycin and monocrotaline, ACE2 has been shown to protect from chronic lung injuries, fibrosis, and pulmonary vasoconstriction [30,31].

It is also pointed out that the decrease in the catalytic function of ACE2 disrupts the RAAS in lungs, resulting in increased inflammation and vascular permeability mediated by ACE/AngII/AT1 axis [32]. On the other hand, the ACE2/Ang1-7/MasR pathway plays an important anti-inflammatory and anti-oxidant role protecting the lungs against ARDS and lethal avian influenza A H5N1 infection. It has been shown that avian influenza A H5N1-infected patients exhibit markedly increased serum levels of Ang II which appear to be linked to the severity and lethality of infection, at least in some cases. Infection with avian influenza A H5N1 virus in experimental mouse models results in the downregulation of ACE2 expression in the lungs and increased serum Ang II levels. Genetic inactivation of ACE2 causes severe lung injury in H5N1-challenged mice, confirming a role of ACE2 in H5N1-induced lung pathologies. Additionally, the application of recombinant human ACE2 (rhACE2) improves avian influenza H5N1 virus-induced lung injury in mice. These results suggest (a) that in acute lung injury, there are decreased ACE2 and increased Ang II levels, and (b) that ACE2 can be supplemented with recombinant ACE2, and Ang II could be inhibited, resulting in improved outcomes [33]. As recent studies in animal models confirmed, ACE2 could be considered as a potential down-regulator for the RAAS. It is also worth mentioning that in addition to the studies on animal models, there are also a certain number of studies conducted on human models of pulmonary arterial hypertension (PAH), which will be discussed later. 

Promoting the ACE2/Ang1-7/MasR signaling by rhACE2 or the Ang 1-7 receptor agonist could also be a promising therapeutic approach in lung disease from diverse etiologies [34]. For example, the Ang 1-7 receptor agonist AVE 0991 has been shown to exert cardio-renal and pulmonary protective effects [35], and treatment with rhACE2 improved the symptoms of acute lung injury, cardiovascular disease, and kidney injury in various preclinical models [36]. Therefore, ACE2 could be considered as a key player in the counteraction of the classical RAAS system because it metabolizes the vasoconstrictive, hypertrophic and proliferative Ang II into the more favorable Ang 1-7 [37]. 

## 3. COPD and ACE2

Chronic obstructive pulmonary disease (COPD) is a systemic disease characterized by lung tissue destruction and progressive airflow limitation that is not fully reversible [38]. COPD affects more than 5% of the world population. According to a study from 2017, the prevalence of COPD increased by 44.2% between 1990 and 2015, while the rate of death in 2015 reached 2.3 million [39]. It is characterized by a chronic cough and progressive dyspnea. The main risk factor is tobacco smoking. Other risk factors, such as genetic predisposition and occupational exposure, contribute to the pathogenesis as well. Chronic inflammation, defined by the infiltration of inflammatory cells and chronic release of proinflammatory cytokines, plays a key role in the pathogenesis of COPD [40,41]. It has been shown that COPD is associated with RAAS activation by means of the action of Ang II, which increases the release of proinflammatory cytokines and promotes the systemic inflammation. More specifically, monocyte chemoattractant protein-1 (MCP-1) derived from the alveolar macrophage has been shown to activate mast cells in response to acute alveolar hypoxia, thus triggering systemic inflammation [42,43]. In addition, it has been observed that Ang II-mediated ROS generation, mitochondrial dysfunction and impaired redox signaling contribute to COPD development [44]. Xue et al. [45] studied the effect of ACE2 expression on lung function and inflammatory response in COPD rat models. They showed that the higher expression of ACE2 in the epithelial cells of the airways is associated with the significant improvement of the lung function and reduced inflammatory cytokine levels, like TNF-α, IL-8, IL-2 and IL-1β. They have also shown that the expression of ACE2 mRNA in the lungs of rats with COPD is markedly decreased, in comparison to wild type, indicating that suppressed ACE2 regulation and ACE/ACE2 imbalance could be associated with COPD pathogenesis and progression [45]. 

The combination of corticosteroids and bronchodilators is a recommended treatment protocol for severe COPD and standard of care in treating the COPD exacerbations. Some of the latest studies have shown that inhaled corticosteroids used in stabilizing a chronic obstructive disease in patients hospitalized for COVID-19 infection decreased the expression of ACE2 receptors [46]. However, the link between inhaled corticosteroids and ACE2 in COPD patients without COVID-19 infection remains unclear. It is well accepted that prolonged exposure to tobacco smoke and environmental irritants are the main risk factors for COPD development. It was suggested that RAAS could be involved in this process in the way that nicotine from the tobacco smoke suppresses the activity of the ACE2-Ang1-7-MasR axis as well as its lung preserving function and increases the activity of the antagonistic ACE/Ang II/AT1R axis, leading to the disruption of the RAAS signaling balance [47].

## 4. Asthma and ACE2

Asthma is a heterogeneous disease of the respiratory tract, characterized by a chronic inflammation of the airways. It is estimated that more than 339 million people worldwide have asthma [48]. Asthma is caused by multiple factors and it depends upon the interaction between a large number of genes and environmental components. Symptoms of the illness include coughing, wheezing, shortness of breath and chest tightness. It is characterized by reversible bronchial hyperactivity of provoking factors, with predominantly eosinophilic infiltrates [49]. However, in severe cases, neutrophilic infiltrates can also be found [50]. The main pathophysiological mechanism in most asthma patients is type 2 inflammation, mediated by Th2 lymphocytes, eosinophils, mastocytes, basophils and IgE–producing plasma cells [51]. The asthma endotype is especially important, as cytokines can modify ACE expression. IL-4 and IL-13 can downregulate ACE2 expression [52], whereas IL-17 can upregulate ACE2 expression [53]. Type 2 inflammation is believed to be the reason for reduced ACE2 gene expression in the airways of patients with allergic asthma [54,55]. A fluctuation of eosinophil levels has been observed in patients with both COVID-19 and asthma. In patients with more severe forms of the infection, the eosinophil level decreases, and upon clinical improvement, it normalizes, suggesting a protective role of eosinophils [54]. Still, there are conflicting opinions, so research on a larger number of subjects is needed [56,57]. The direct role of ACE2 in asthma has not yet been proved, but the presumed model suggests a role of ACE2 in inactivating the AngII/AT1R and Ang1-7/MasR axes. Thus far, it is clear that AngII-AT1R binding causes bronchoconstriction and that AT1R antagonists prevent eosinophilic infiltration and antigen-induced airway hyperactivity in guinea pigs [58]. It has also been proved that Ang 1-7 is able to reduce the symptoms of allergic asthma with its anti-inflammatory properties [59,60]. According to this, Ang 1-7, as a RAAS component, could be involved in the progression of asthma. It is known that airway remodeling, as a consequence of exacerbation, aggravates the airway obstruction and accelerates poor disease outcome [61]. In order to achieve optimal disease control, molecular studies focused on the identification of specific genes and their interacting partners are of special importance [62]. Due to the fact that AT1R antagonists and Ang 1-7 proved to be promising in preventing eosinophilic infiltrates and airway hyperactivity in animal models [58], as well as moderating allergic asthma [59], it is to be expected that ACE2 will find its place in the pathophysiology of asthma, given the fact that ACE2 levels in the sputum of asthma patients are greater than those of the healthy population [63].

## 5. Pulmonary Hypertension and ACE2

Pulmonary hypertension (PH) is a chronic progressive disease defined by the mean pulmonary arterial pressure (mPAP) > 20 mmHg, measured during right heart catheterization [64]. The key processes in the pathophysiology of pulmonary hypertension are the dysfunction of arterial endothelial cells in the lungs, inflammation, vasoconstriction, thrombosis and abnormal vascular proliferation. PH can be divided into five groups, according to the clinical presentation and hemodynamic features [64]. Even though pulmonary arterial hypertension (PAH) is a rare disease with a prevalence of 15–60 patients per million and an incidence of 5–10 cases per million, it is the only group with a disease-specific therapy [65]. Here, we will present the current knowledge of PAH, followed by the discussion of the potential role of RAAS components in the PH pathogenesis. Some of the major elements in PAH pathogenesis are mitochondrial dysfunction, altered expression and function of certain growth factors, and the disturbed regulation of the immune cells, such as B-and T-lymphocytes, mast cells, dendritic cells and macrophages [66]. The excessive local secretion of IL-1, IL-6, LTB4, macrophage migration inhibitory factor, leptin and TNF-α, and the inactivation of FoxO1 play an integral role in mediating the structural and functional changes in the pulmonary vasculature in PAH. Impaired T-regulatory cell function, T helper 17 cell immune polarization [67] and dendritic cell recruitment in pulmonary vascular lesions have been demonstrated in tissues from PAH patients, supporting dysregulated immune response [66]. The existing knowledge of the RAAS and its role in the pathogenesis of PH is based on the model in which Ang II-induced vasoconstriction, endothelial cell proliferation and inflammation promote the development of disease [68]. The development of PH, in experimental models, was slowed by suppressive activity of exogenous ACE2 and Ang 1-7 on Ang II, suggesting that ACE2 could have a protective role in the progression of PH [69,70]. As shown by the in vivo studies, thickening of the walls of pulmonary arteries is inhibited by the overexpression of ACE2 [31]. Additionally, it has been shown in the animal models of tobacco smoke-induced PAH that the ACE2 activity was downregulated [71]. Studies conducted on patients with PAH reported decreased levels of serum ACE2 and Ang 1-7 [72], which was consistent with findings of Zang et al., who demonstrated decreased expression and phosphorylation of ACE2 in remodeled pulmonary arteries in explanted lungs from patients with idiopathic PAH [73]. Therefore, it is realistic to expect that a potential ACE2 activator could be used in the treatment of patients with PH and in the prevention of the disease progression. The available medications are indicated only for specific types of pulmonary hypertension: PAH and chronic thromboembolic pulmonary hypertension (CTEPH) [65]. In general, better understanding of PH pathophysiology would enable earlier identification of persons at risk, earlier diagnosis and, consequentially, an earlier start of the specific treatment.

## 6. COVID-19 and ACE2

The COVID-19 pandemic has restored the interest in ACE2 because it serves as a receptor for SARS-CoV-2 virus entry into the cell [74]. In the beginning of the 2020, the COVID-19 disease was declared a pandemic by the World Health Organization [75]. The new disease began as a seemingly ordinary respiratory infection, confined to the Chinese city of Wuhan, only to expand globally in a matter of months [76]. The newly characterized coronavirus was named SARS-CoV-2 [77]. Mild clinical presentations affect the upper airways. Patients with moderate disease have pneumonia which can, in the most severe cases, progress to acute respiratory failure requiring mechanical ventilation. Epidemiological studies have shown that the patients with the severe form of COVID-19 had other comorbidities, including arterial hypertension and other cardiovascular diseases, diabetes, cerebrovascular diseases, and chronic lung diseases, were overweight and were older than 60 [78,79]. COPD has been proven to be one of the major risk factors for both a severe infection and an adverse outcome [80]. There are still many unknown facts about the pathophysiology of the virus and the body’s immune response to it, but according to recent studies, ACE2 has a role in explaining some of these processes [81]. ACE2 has been indicated as a receptor for the SARS-CoV2 spike protein (S). The work by Kuba et al. in a mouse model of acute lung injury induced by the SARS-CoV spike protein suggested that the spike protein binds to ACE2 and subsequently downregulates ACE2 protein expression, leading to worsening of the ARDS symptoms. They also demonstrated that the mechanism underlying this process is the enhanced vascular permeability by elevated Ang II level. Therefore, they proposed that ACE2 and other components of the RAAS pathway might play a pivotal role in the SARS pathogenesis during progression to ARDS [82]. It has also been shown that application of the recombinant (rACE2) speeds up the recovery of lung failure [12]. It was observed that patients with COVID-19 show an imbalance in the RAAS system’s activity due to the loss of the ACE2, which further contributes to tissue and systemic inflammation [83,84]. For example, experimental animal models using lipopolysaccharide (LPS)-induced acute lung injury have shown decreased expression of ACE2, increased inflammatory injury, and upregulated expression of renin, Ang II, ACE, and AT1R. After the injection of rhACE2, lung function and pathological injury were improved, followed by attenuated inflammation [85].

## 7. Conclusions

The incidence and prevalence rates of chronic inflammatory lung diseases are increasing despite all the available medications and therapies. Given the fact that the majority of the described models of ACE2 receptor involvement in chronic inflammatory lung diseases came from animal models of acute inflammatory lung diseases, it is necessary to conduct more studies on humans, aiming to gain better insight into the pathogenesis of the disease. Our research group is largely focused on the genetic diversity of different genes involved in the innate immune response in patients with COPD and lung cancer, and functional studies of disease-associated variants. We would be very interested to see how specific genetic variants of ACE2, in these groups of patients, affect protein function and regulation of the RAAS, and whether these changes are potentially responsible for the pathological condition. It is also worth mentioning that ACE2 was analyzed as a potential biomarker for SARS-CoV-2 infection risk in lung cancer patients. It seems that upregulated expression of ACE2 in lung tumors might increase the susceptibility to COVID-19 infection [86]. On the other hand, Feng et al. found that ACE2 overexpression, regardless of SARS-CoV-2 infection, may have a protective effect by inhibiting cell growth and VEGEF production [87]. Therefore, it seems to us that ACE2 could be considered as a potential biomarker of inflammatory lung diseases or even a standard component in the medical treatment of COPD, asthma or pulmonary hypertension.

## Figures and Tables

**Figure 1 genes-12-01054-f001:**
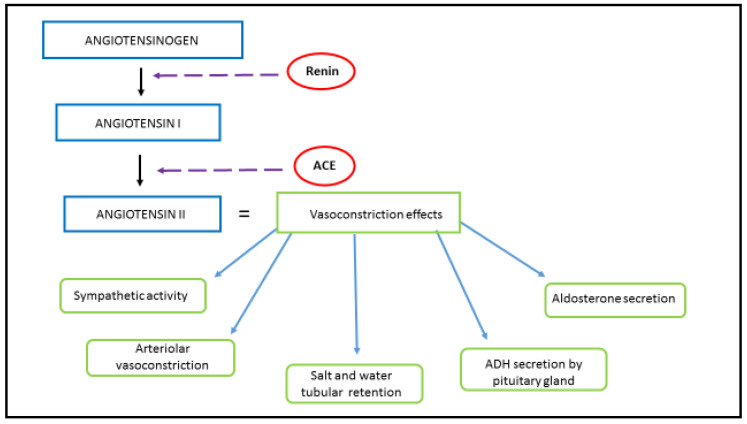
RAAS and Ang II. Renin, the proteolytic enzyme, is secreted from the juxtaglomerular cells of the kidney as a response to its hypoperfusion. Renin hydrolyzes angiotensinogen, secreted by the liver, into Ang I. Ang I has a weak biological effect until converted into Ang I via ACE, which is produced in the lungs. Schematic representation of Ang II origin and its vasoconstrictive effect, achieved by acting on the autonomic nervous system, renal and systemic blood vessels, renal tubules, adrenal glands and the posterior pituitary lobe.

**Figure 2 genes-12-01054-f002:**
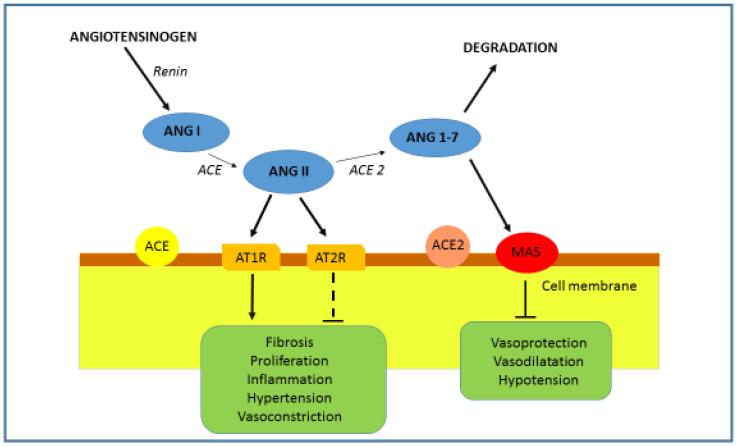
RAAS and ACE2. The role of ACE2 is the degradation of Ang II into Ang 1-7. The binding of Ang 1-7 on the Mas R results in an effect opposite to that of Ang II. By acting on the Mas R vasodilatation, anti-inflammatory, antihypertrophic, antiproliferative and antifibrotic effects are promoted.

## Data Availability

Not applicable.

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
