# Peer review of "Angiotensin-Converting Enzyme 2 (ACE2) as a Potential Diagnostic and Prognostic Biomarker for Chronic Inflammatory Lung Diseases"

_genes, 2021, doi:10.3390/genes12071054_

Round 1
Reviewer 1 Report
Firstly, I would thank for giving me the important chance to read the submitted manuscript, which was written comprehensively well. Accordingly, Angiotensin-Converting Enzyme 2 (ACE2), an important component of the Renin-Angiotensin-Al-dosterone System (RAAS), is expressed in the airways. And, ACE2 could play as a role in the systemic regulation of cardiovascular and renal systems, lungs and liver by acting on blood pressure, electrolyte balance control mechanisms and inflammation. The authors had systemically reviewed and discussed many reports about ACE2, including its potential role in the pathophysiology of the chronic inflammatory lung diseases: COPD, asthma and pulmonary hypertension over the last 20 years since its discovery. In this manuscript, except the recent findings that ACE2 has multiplicity of physiological roles that involve around its trivalent function, such as a negative regulator of RAS system, facilitator of amino acid tansport , and SARS-CoV-2 receptor recently, there are few human studies directly supporting that ACE2 can be closely involved in the pathogenesis of inflammatory lung diseases. Therefore, I would suggest as follows, 1) To add some human studies with inflammatory lung diseases not related to COVID-19, including pharmacologic therapy or preventive strategy with rhACE2 or ACE2 activator activators, in stead of experimental animal models. 2) To delineate how ACE2 is related to vasomediators (NO , endothelin-1 and prostaglandin), mitochondrial dysfunction or oxidation in inflammatory lung disease, especially pulmonary hypertension. 3) To discuss the relationship between cancer biomolecular behavior and ACE2 in inflammatory lung diseases.
Author Response
1) To add some human studies with inflammatory lung diseases not related to COVID-19, including pharmacologic therapy or preventive strategy with rhACE2 or ACE2 activator activators, instead of experimental animal models.
We understand the Reviewer's concern and we add relatively recent data related to human studies conducted on PAH and ACE2. It is important to say that it is not easy to find studies conducted on human models that are not related to COVID-19. We hope that in this way we, at least partially, met the Reviewers requirements.
2) To delineate how ACE2 is related to vasomediators (NO, endothelin-1 and prostaglandin), mitochondrial dysfunction or oxidation in inflammatory lung disease, especially pulmonary hypertension.
We understand and appreciate Reviewer's comment, however, it seems to us that majority of suggested topics are briefly discussed in the current version of the manuscript, especially in the part where we elaborate role of ACE2 in pulmonary hypertension. We hope that the Reviewer will reconsider his decision in a positive way and accept the current form.
3) To discuss the relationship between cancer biomolecular behaviour and ACE2 in inflammatory lung diseases.
We agree with Reviewer that it will be interesting to comment on the relationship between lung cancer, inflammatory lung diseases, and ACE2 as a potential biomarker. In dead, there are already some published data on ACE2 expression in non-small cell lung cancer (NSCLC) showing that ACE2 expression is decreased in tumour tissues, when compared with non-malignant tissue, indicating that this data could be considered as a new strategy in NSCLC treatment (DOI: 10.3892/or_00000718; ONCOLOGY REPORTS 23: 941-948, 2010). Recent analyses are mainly focused on lung cancer and the risk of SARS-CoV-2 infection. It seems that this is a rather complex area and beyond the scope of this research. Therefore we only mentioned in the Conclusion part of the manuscript that the relationship between cancer and ACE2 as a potential biomarker should be considered. We hope that in this way we met the Reviewers requirements.
Reviewer 2 Report
The review by Marčetić et al. focuses on the link between the ACE2 and airway inflammation. This is a concise, informative and timely manuscript, and I only have a few minor comments:
- Page 3 line 1: “ACE2 has a multiplicity of physiological roles that revolve around its dual function” the exploit of the Sars-Cov-2 virus to use ACE2 as a binding receptor cannot be referred to as “physiological role” of the receptor.
- Page 4 line 16: The publication by Imai et al is missing in the reference list.
- Page 5 line 33: The publication by Tian et al is missing in the reference list.
- The introduction is densely written and quite long. Dividing it by sub-headings would help the reader.
Author Response
REVIEWER 2 – Answer to the comments
The review by Marčetić et al. focuses on the link between the ACE2 and airway inflammation. This is a concise, informative and timely manuscript, and I only have a few minor comments:
1) Page 3 line 1: “ACE2 has a multiplicity of physiological roles that revolve around its dual function” the exploit of the Sars-Cov-2 virus to use ACE2 as a binding receptor cannot be referred to as “physiological role” of the receptor.
We understand Reviewer concern and changed the text as follows: ACE2 has different roles ranging from a negative regulator of the renin-angiotensin system (peptidase-dependent), an amino acid transporter, and as a functional receptor for severe acute respiratory syndrome coronavirus (SARS-CoV) (peptidase-independent). We hope that this form will be more appropriate. It seems to us that it is important to mention that one of the many roles of ACE2 is its function as a receptor for SARS-CoV. We hope that in this way we met the Reviewers requirements.
2) Page 4 line 16: The publication by Imai et al is missing in the reference list; Page 5 line 33: The publication by Tian et al is missing in the reference list.
We accepted the Reviewer comments and correct the references accordingly.
3) The introduction is densely written and quite long. Dividing it by sub-headings would help the reader.
We agree with Reviewer and in the new version of the manuscript introduction part of the text is divided in the sub-headings. We hope that the new version of the Introduction will fulfil the Reviewers requirements.
Reviewer 3 Report
Angiotensin-converting enzyme 2 (ACE2), is a member of the renin-angiotensin system (RAS) which regulates the respiratory system and cardiovascular system. Researchers found ACE2 highly relevant to lung diseases. It is novel that the authors demonstrated the role of ACE2 in lung inflammatory diseases. However, there are some concerns the authors need to be addressed.
- The title indicates the inflammatory lung diseases which means acute and chronic diseases. However, in abstract, first sentence implies the authors would like to focus on chronic inflammatory lung diseases. Besides, the part 2. the authors described the acute lung inflammatory diseases, ARDS, etc. So, the authors should make the topic consistent.
- As a review, the authors should not only list the published results but also need to summarize by themselves. Thus, in this paper, it is better to generate a table listed ACE2 function, in which disease, mechanism (though which pathway), human or mouse model, reference, etc.
- The authors put lot of effort to explain what are COPD, PH, asthma and COVID-19 rather than discuss the link between the mechanism of ACE2 and those diseases.
- In discussion, the authors stated “Given the fact that all described/suggested models of ACE2 receptor involvement in inflammatory lung diseases came from animals, it is necessary to conduct the studies on humans, aiming to get better insight into the pathogenesis of the disease.” which is not accurate. For example, there is a clinical trial which resulted in that ACE2 blockers did not change the live ratio in mild to moderate COVID-19 patients. (JAMA. 2021;325(3):254-264. doi:10.1001/jama.2020.25864).
- To sum up, the authors need to re-model this manuscript focusing on the topic, and need more representative references to support your idea, and discuss more deeply.
Author Response
REVIEWER 3 – Answer to the comments
Angiotensin-converting enzyme 2 (ACE2), is a member of the renin-angiotensin system (RAS) which regulates the respiratory system and cardiovascular system. Researchers found ACE2 highly relevant to lung diseases. It is novel that the authors demonstrated the role of ACE2 in lung inflammatory diseases. However, there are some concerns the authors need to be addressed.
1) The title indicates the inflammatory lung diseases which means acute and chronic diseases. However, in abstract, first sentence implies the authors would like to focus on chronic inflammatory lung diseases. Besides, the part 2. the authors described the acute lung inflammatory diseases, ARDS, etc. So, the authors should make the topic consistent.
We agree with Reviewer notes regarding the chronic and acute forms of inflammatory lung diseases and in the new version of the manuscript, we tried to unify the terminology and tried to be more consistent. We hope that in this way we met the Reviewers requirements.
2) As a review, the authors should not only list the published results but also need to summarize by themselves. Thus, in this paper, it is better to generate a table listed ACE2 function, in which disease, mechanism (though which pathway), human or mouse model, reference, etc. To sum up, the authors need to re-model this manuscript focusing on the topic, and need more representative references to support your idea, and discuss more deeply.
We understand Reviewer's comments and we agree that table presentation would possibly be more informative. However, we think that the current way of describing different pathologies is not cardinally bad either. Also due to the lack of the given time for response to Reviewers comments, we thought that it can remain in this form.
Regarding the Reviewer comment that more representative references should be used, we add some more recent references and hope that in this way we met reviewer requirements. Regarding the Reviewer's comment that we should summarize the published results by our self, we add some more original views in the Conclusion part of the manuscript. We hope that the new form of the manuscript will be in accordance with Reviewer's expectations.
3) The authors put lot of effort to explain what are COPD, PH, asthma and COVID-19 rather than discuss the link between the mechanism of ACE2 and those diseases.
We understand Reviewer's concern, however have to say that only partially agree with that comment. We agree that some descriptive parts of the text related to an extensive description of the disease's clinical status could be deleted and we did so.
We also understand the Reviewer's comment that the link between the mechanism of ACE2 activation/regulation and disease conditions could be better explained. However, it seems to us that the link between the mechanism of ACE2 and selected disease conditions is relatively well explained. In the introduction part of the manuscript, we explained the physiology and signaling pathway of the RAAS, and the function of the ACE2 in the context of this pathway. Further on, we focused our research on the function of ACE2 in inflammation and pulmonary diseases in general where we discussed the potential protective role of ACE2 described so far in animal models. Finally, we reviewed some of the published data related to selected chronic pulmonary disease, together with the role of ACE2 in the context of COVID-19. It seems to us that in all the above-mentioned parts of the text we have touched on the connection between the mechanism of action of ACE2 and pathological conditions. We are aware that it can always be better, but we hope that the Reviewer will reconsider his decision in a positive way and accept the current form.
4) In discussion, the authors stated “Given the fact that all described/suggested models of ACE2 receptor involvement in inflammatory lung diseases came from animals, it is necessary to conduct the studies on humans, aiming to get better insight into the pathogenesis of the disease.” which is not accurate. For example, there is a clinical trial which resulted in that ACE2 blockers did not change the live ratio in mild to moderate COVID-19 patients. (JAMA. 2021;325(3):254-264. doi:10.1001/jama.2020.25864).
We agree with the Reviewer’s opinion that the above-mentioned sentence is slightly strange formulated and therefore in the new version of the manuscript it is changed: Given the fact that the majority of the described models of ACE2 receptor involvement in chronic inflammatory lung diseases came from animal models of acute inflammatory lung diseases, it is necessary to conduct more studies on humans, aiming to get better insight into the pathogenesis of the disease.
Regarding the Reviewer's comment that it is not accurate, it seems to us that it is a misunderstanding. The above-mentioned paper, suggested by the Reviewer, is a clinical trial related to the ACEIs and ARBs. The aim of the study was to determine how intake of those drugs is related to the duration of hospitalization. It is interesting to study, however, the aim of our research was not related to that. We hope that additional references included in the new version of the manuscript will meet the Reviewers requirements.
Round 2
Reviewer 3 Report
The authors have addressed my concerns.